

# Phosphoserine for the generation of lanthanide binding sites on proteins for paramagnetic NMR

Sreelakshmi Mekkattu Tharayil[1,*], Mithun Chamikara Mahawaththa[1,*], Choy-Theng Loh[1,2], Ibidolapo Adekoya[1], Gottfried Otting[1]

[1] ARC Centre of Excellence for Innovations in Peptide and Protein Science, Research School of Chemistry, Australian National University, Canberra ACT 2601, Australia

[2] present address: Hangzhou Wayland Bioscience Co. Ltd, Hangzhou 310030, PR China

[*] The first two authors contributed equally.

*Correspondence to*: Gottfried Otting (gottfried.otting@anu.edu.au)

**Abstract.** Pseudocontact shifts (PCS) generated by paramagnetic lanthanide ions provide valuable long-range structural information in NMR spectroscopic analyses of biological macromolecules such as proteins, but labelling proteins site-specifically with a single lanthanide ion remains an ongoing challenge, especially for proteins that are not suitable for ligation with cysteine-reactive lanthanide complexes. We show that a specific lanthanide binding site can be installed on proteins by incorporation of phosphoserine in conjunction with other negatively charged residues, such as aspartate, glutamate or a second phosphoserine residue. The close proximity of the binding sites to the protein backbone leads to good immobilization of the lanthanide ion, as evidenced by the excellent quality of fits between experimental PCSs and PCSs calculated with a single magnetic susceptibility anisotropy ($\Delta\chi$) tensor. An improved two-plasmid system was designed to enhance the yields of proteins with genetically encoded phosphoserine and good lanthanide ion affinities were obtained when the side chains of the phosphoserine and aspartate residues are not engaged in salt bridges, although the presence of too many negatively charged residues in close proximity can also lead to unfolding of the protein. In view of the quality of the $\Delta\chi$ tensors that can be obtained from lanthanide binding sites generated by site-specific incorporation of phosphoserine, this method presents an attractive tool for generating PCSs in stable proteins, particularly as it is independent of cysteine residues.

## 1 Introduction

Paramagnetic labels offer an attractive tool for the study of protein structure and function, as the magnetic moments of unpaired electrons generate long-range paramagnetic effects in NMR spectra. Among the paramagnetic effects that can be observed in NMR spectra, pseudocontact shifts (PCS) generated by paramagnetic metal ions stand out for their high information content and ease of observation (Otting, 2008; Parigi and Luchinat, 2018). Specifically, the PCSs provide information about the location of nuclear spins relative to the magnetic susceptibility anisotropy tensor ($\Delta\chi$ tensor) associated with a paramagnetic metal ion, and this information can readily be obtained for nuclear spins as far as 40 Å from the paramagnetic centre (Bertini et al., 2001).





As lanthanide ions display particularly large $\Delta\chi$ tensors (Bleaney, 1972; Bertini et al., 2001), significant efforts have been made to devise lanthanide complexes for site-specific tagging of proteins (Su and Otting, 2010; Keizers and Ubbink, 2011; Nitsche and Otting, 2017; Joss and Häussinger, 2019; Saio and Ishimori, 2020). In an alternative approach, PCSs can be elicited in proteins by creating binding sites for lanthanides or lanthanide complexes by protein engineering (Yagi et al.,
2010; Barthelmes et al., 2011, 2015; Jia et al., 2011).

A common problem of lanthanide tags arises from mobility of the metal-ion complex relative to the target protein. While paramagnetic lanthanide ions generate paramagnetic relaxation enhancements (PRE) in the protein irrespective of metal mobility, PCSs can decrease dramatically if the lanthanide complex reorientates relative to the protein. With a limited degree of tag flexibility, the PCSs may still be explained by a single effective $\Delta\chi$ tensor although, in principle, a family of $\Delta\chi$ tensors
would be required to account for multiple tag conformations (Shishmarev and Otting, 2013). Well immobilized metal ions thus not only deliver larger PCSs but also more reliable $\Delta\chi$-tensor fits.

Different strategies have been devised to immobilise lanthanide ions on proteins. Tag motions can be restricted by short tethers and bulky lanthanide complexes to hem in the tag sterically (Nitsche and Otting, 2017). Double-arm tags provide two attachment points (Keizers and Ubbink, 2011), but even these designs have shown signs of tag mobility (Hass et al., 2010).
A lanthanide-binding peptide (LBP) engineered into polypeptide loops of protein structures can deliver good metal immobilization but presents a major modification of the target protein (Barthelmes et al., 2011; 2017). Fusions of an LBP combined with disulfide bond formation have also been explored, but do not necessarily achieve good immobilisation of the lanthanide ion (Saio et al., 2009, 2010, 2011). A successful strategy has been a design, where two neighbouring cysteine residues are furnished with metal chelating tags and a single lanthanide ion is coordinated by both chelating groups (Swarbrick
et al., 2011; Welegedara et al., 2017), a design that has also proven successful for $Co^{2+}$ ions (Swarbrick et al., 2016). The most serious drawback of this design is its reliance on cysteine residues, which makes it incompatible with proteins that contain functionally important cysteine residues in their wild-type sequence. In fact, most of the currently available lanthanide tags target cysteines (Su and Otting, 2010; Keizers and Ubbink, 2011; Nitsche and Otting, 2017; Joss and Häussinger, 2019; Saio and Ishimori, 2020), as thiol groups can readily undergo selective chemical reactions. To avoid the mobility of solvent-exposed
cysteine side chains, tags have also been designed for attachment to the side chains of aromatic residues, which are more hindered sterically and thus discouraged from populating different rotamers (Loh et al., 2015; Abdelkader et al., 2016), but this approach results in long linkers between the lanthanide ion and the protein, increasing the chances that the lanthanide ion moves and reorientates relative to the protein backbone.

The most elegant strategy for generating a lanthanide binding site in a protein would be to introduce a lanthanide-
binding unnatural amino acid that can be site-specifically incorporated by genetic encoding. This approach would relieve any reliance on cysteine residues. Although systems for genetic encoding have been devised for over 100 different unnatural amino acids, only few of these can bind metal ions (Dumas et al., 2014) and those that do were found to precipitate proteins upon binding lanthanide ions. For example, protein precipitation has been reported for 2-amino-3-(8-hydroxyquinolin-3-yl) propanoic acid (HQ-Ala; Jones et al., 2009) and we found ourselves incapable of improving on these results. Similarly,



bipyridyl-alanine (Bpa) was shown to allow binding of $Co^{2+}$ and the observation of PCSs (Nguyen et al., 2011), but subsequent experiments with Bpa incorporated in different proteins and at different sites showed that also this system is prone to precipitating proteins upon addition of metal ion.

In the present work, we explored the potential of a different unnatural amino acid, phosphoserine (Sep), to create a lanthanide binding site. Lanthanide ions are known for their affinity to negatively charged oxygens and, with a $pK_a$ value of

5.6 for the equilibrium between monobasic and dibasic forms (Xie et al., 2005), a phosphoserine residue carries two negative charges under physiological conditions. Phosphorylation of serine residues is a well-known posttranslational modification of proteins effected by kinases, but this often is neither quantitative nor easily achievable for specific serine residues. Recently, however, an orthogonal phosphoseryl-tRNA-synthetase/tRNA pair has become available, which allows installing a Sep residue in response to an amber stop codon (Lee et al., 2013; Pirman et al., 2015; Yang et al., 2016). In the following we show that the

system is sufficiently effective to install two Sep residues in the same protein, explore the potential to create a lanthanide binding site using a Sep residue in conjunction with other negatively charged residues, in particular an aspartate or a second Sep residue, and demonstrate the exceptional quality of $\Delta\chi$ tensors that can be obtained with lanthanide ions in these sites.

## 2 Experimental procedures

### 2.1 Plasmid preparation for protein expression

The plasmid SepOTSλ, which contains the phosphoseryl-tRNA synthetase/tRNA pair and a suitable EF-Tu mutant for incorporation of Sep in response to an amber stop codon (Pirman et al., 2015), was obtained from Addgene. To create a T7 expression vector that is compatible with SepOTSλ, we subcloned the region containing the T7 promoter, ribosome binding site, multiple cloning site and T7 terminator from pETMCSIII (Neylon et al., 2000) into the plasmid pCDF (Lammers et al., 2014). The gene of interest was inserted into the multiple cloning site and furnished with a C-terminal His$_6$-tag preceded by a

TEV cleavage site. All plasmid constructions were conducted with a QuikChange protocol using mutant T4 DNA polymerase (Qi and Otting, 2019).

### 2.2 Protein expression

All proteins were expressed in the BL21ΔserB strain (Park et al., 2011), which lacks phosphoserine phosphatase and thus minimizes the dephosphorylation of phosphoserine to serine. The SepOTSλ and pCDF plasmids were co-transformed into

electrocompetent BL21ΔserB cells. In order to minimize usage of amino acids and $^{15}NH_4Cl$, the following top-down expression method was used. Initially, 1 litre of cell-culture was grown in LB medium with 25 μM spectinomycin and 20 μM kanamycin at 37 °C until the $OD_{600}$ value reached 0.6–0.8. Next, the cells were pelleted and resuspended in 300 mL M9 medium (6 g L$^{-1}$ $Na_2HPO_4$, 3 g L$^{-1}$ $KH_2PO_4$, 0.5 g L$^{-1}$ NaCl) and supplied with 1 g L$^{-1}$ $^{15}NH_4Cl$ and 1 mM phosphoserine. Subsequently, the



cells were incubated for 30 minutes at 37 °C and induced with IPTG. Protein expression was conducted at 25 °C overnight.

Cells were harvested by centrifugation at 5,000 $g$ for 15 minutes and lysed by passing twice through a French Press (SLM Amicon, USA) at 830 bars. The lysate was then centrifuged at 13,000 $g$ for 60 minutes and the filtered supernatant was loaded onto a 5 mL Ni-NTA column (GE Healthcare, USA) equilibrated with binding buffer (50 mM Tris-HCl, pH 7.5, 300 mM NaCl, 5 % glycerol). The protein was eluted with elution buffer (binding buffer containing, in addition, 300 mM imidazole) and the fractions were analysed by 12% SDS-PAGE. For the double-amber mutants, the His$_6$-tag was removed by digestion

overnight at 4 °C, using TEV protease added in 100-fold excess in buffer containing 50 mM Tris-HCl, pH 8.0, 300 mM NaCl and 1 mM β-mercaptoethanol. The resulting protein samples were then treated with 5 mM EDTA to remove any di- or trivalent metal ion that could have been adsorbed during protein expression and purification. Finally, EDTA was removed by buffer exchange with NMR buffer (20 mM HEPES-KOH, pH 7.0) using a HiPrep desalting column (GE Healthcare, USA). Mass-spectrometric analysis was conducted using an Elite Hybrid Ion Trap-Orbitrap mass spectrometer (Thermo Scientific, USA)

coupled with an UltiMate S4 3000 UHPLC (Thermo Scientific, USA). 7.5 pmol of sample were injected to the mass analyser via an Agilent ZORBAX SB-C3 Rapid Resolution HT Threaded Column (Agilent, USA).

## 2.3 NMR spectroscopy

All NMR spectra were recorded at 25 °C, using an 800 MHz Bruker Advance NMR spectrometer for all mutants containing a single phosphoserine residue and a 600 MHz Bruker Advance NMR spectrometer for all mutants containing two phosphoserine

residues. Samples were prepared in 20 mM HEPES buffer, pH 7.0, in 3 mm NMR tubes. 10 % D$_2$O was added to provide a lock signal. 0.1– 0.5 mM protein samples were used for 2D [$^{15}$N,$^1$H]-HSQC experiments. Complexes with lanthanides were obtained by titration with 10 mM LnCl$_3$ stock solutions.

## 2.4 PCS measurements and Δχ-tensor fitting

Pseudocontact shifts (PCS) were measured in ppm as the difference in amide proton chemical shift between the paramagnetic

and diamagnetic NMR spectrum. PCSs were used to determine the position and orientation of the Δχ-tensor of the paramagnetic ions relative to the protein structure. Fitting of Δχ tensors was performed using the program Paramagpy (Orton et al., 2020).

## 2.5 Isothermal titration calorimetry

Isothermal calorimetric titration experiments were performed using a Nano-ITC low volume calorimeter (TA Instruments, USA) at 25 °C with stirring at 250 rpm. The protein mutant E18Sep and the titrants TbCl$_3$ and TmCl$_3$ were prepared in the same buffer (20 mM HEPES, pH 7.0) and degassed before use. Data were analysed using the programs NITPIC and SEDPHAT



(Keller et al., 2012). The baseline-subtracted power peaks were integrated, and the integrated heat values fitted to the single binding site model (A + B ↔ AB, heteroassociation) to obtain the dissociation constant ($K_d$). The global fitting was done by

repeatedly cycling between Marquardt–Levenberg and Simplex algorithms in SEDPHAT until modelling parameters converged; 68 % confidence intervals were calculated using the automatic confidence interval search with the projection method using F-statistics in SEDPHAT.

# 3 Results

## 3.1 Phosphoserine incorporation

Simultaneous transfection of *E. coli* with the SepOTSλ plasmid and pET vectors containing the genes of proteins targeted for overexpression and Sep incorporation led to slow cell growth and variable colony sizes on plates as described earlier (Pirman et al., 2015). Noting that the SepOTSλ plasmid contains the origin of replication of pUC, which belongs to the same plasmid incompatibility group as pET vectors (Morgan, 2014), we constructed a new expression vector based on pCDF to include T7 promoter, ribosome binding site, multiple cloning site and T7 terminator. This modification restored the usual growth rates of

the cells. Proteins containing phosphoserine were expressed from a two-plasmid system containing SepOTSλ and a modified pCDF vector in BL21ΔserB. Expression yields of up to 3 mg purified protein per litre of growth medium were obtained.

## 3.2 Single phosphoserine residues for lanthanide binding

We used the proteins ubiquitin and GB1 to test whether a single phosphoserine residue is sufficient to create a lanthanide binding site. We hypothesized that a phosphoserine residue assisted by an additional carboxy group from a glutamate or

aspartate residue (in the following referred to as 'helper residue') could potentially be sufficient to generate a tridentate complex with a lanthanide ion, positioning the metal ion close to the protein backbone and compensating its positive charge. In the first example, we made the mutant E18Sep of ubiquitin, where E16 and D21 could act as potential helper residues. Subsequent titration with $Tb^{3+}$ ions succeeded in generating PCSs of up to almost 1 ppm (Table S1). The paramagnetic peaks appeared at chemical shifts different from the diamagnetic parent peaks, indicating slow exchange between lanthanide-bound

and free protein. Isothermal calorimetric experiments with $Tb^{3+}$ and $Tm^{3+}$ ions indicated dissociation constants of 45 and 33 μM, respectively (Fig. S1).

Figure 1a shows the PCSs observed with $Tb^{3+}$ and $Tm^{3+}$ ions after addition in equimolar ratio. Using the NMR ensemble structure of ubiquitin (PDB ID: 2KOX; Fenwick et al., 2011) and the measured PCSs, the metal position was determined by fitting the Δχ tensor using the program Paramagpy (Table 1; Orton et al., 2020). The correlation between back-

calculated and experimental PCSs was excellent (Fig. 2a, Table S1), resulting in a *Q*-factor of 0.03. This indicated that the Sep residue and lanthanide complex did not alter the structure of the protein. Furthermore, the tensor fit positioned the lanthanide



ion between the phosphoserine residue and D21, suggesting that D21 acts as a helper residue rather than E16. To verify this result, we prepared the two ubiquitin mutants E18Sep/E16Q and E18Sep/D21N. As expected, the former delivered similar PCSs (Fig. 1b, Table S1), a similarly good $\Delta\chi$-tensor fit (Table 1) and a similar metal position, whereas the latter showed only

very small chemical shift changes upon titration with lanthanides, indicating a faster exchange (Fig. 1c). The paramagnetic centre identified by the fits placed the lanthanide ions between the aspartate and Sep residues as expected (Fig. 2b).





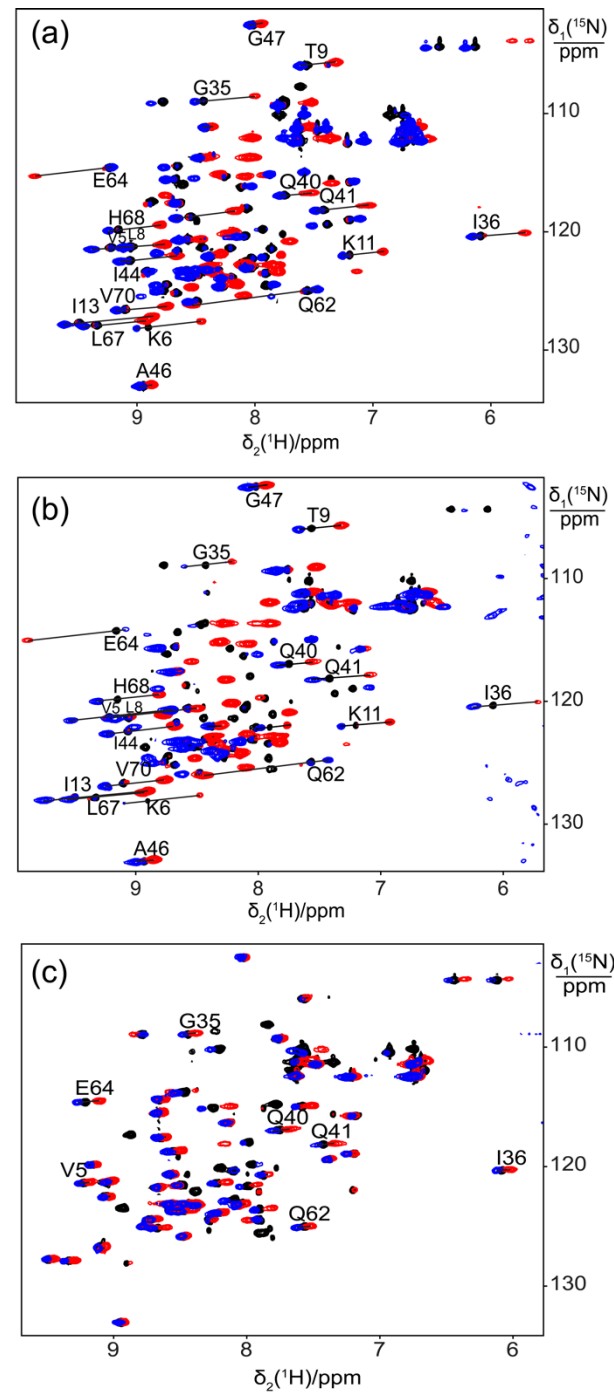

**Figure 1.** Superimposition of [$^{15}$N,$^{1}$H]-HSQC spectra of 0.5 mM solutions of $^{15}$N-labelled ubiquitin mutated to generate a
lanthanide binding site at residue 18. Spectra with diamagnetic Y$^{3+}$ are plotted in black and with paramagnetic Tb$^{3+}$ and Tm$^{3+}$





in red and blue, respectively. Lines were drawn to connect some of the cross-peaks belonging to the same residue in the paramagnetic and diamagnetic samples and are labelled with the residue name and sequence number. (a) Mutant E18Sep. (b) E16Q/E18Sep. (c) E18Sep/D21N.


**Table 1.** $\Delta\chi$-tensor parameters of the ubiquitin mutants E18Sep, E16Q/E18Sep and T22Sep/N25D/K29Q and the GB1 mutant K10D/T11Sep complexed with $Tb^{3+}$ and $Tm^{3+}$ ions.[a]

| Protein | $\Delta\chi_{ax}$[b] $(10^{-32}\ m^3)$ | $\Delta\chi_{rh}$[b] $(10^{-32}\ m^3)$ | x (Å) | y (Å) | z (Å) | $\alpha$ (°) | $\beta$ (°) | $\gamma$ (°) | $Q$[c] |
|---|---|---|---|---|---|---|---|---|---|
| ubiquitin E18Sep ($Tb^{3+}$) | 17.1 (0.6) | 2.8 (0.3) | 10.095 | -1.846 | -11.711 | 170 | 138 | 50 | 0.03 |
| ubiquitin E18Sep ($Tm^{3+}$) | -2.7 (0.1) | -1.0 (0.1) | 9.463 | -0.674 | -12.207 | 168 | 129 | 49 | 0.03 |
| ubiquitin E16Q/E18Sep ($Tb^{3+}$) | 15.9 (0.6) | 3.4 (0.8) | 9.695 | -1.754 | -11.833 | 162 | 135 | 37 | 0.03 |
| ubiquitin E16Q/E18Sep ($Tm^{3+}$) | -4.5 (0.1) | -2.1 (0.1) | 9.441 | -1.902 | -11.918 | 164 | 131 | 59 | 0.03 |
| GB1 K10D/T11Sep ($Tb^{3+}$) | 27.9 (0.1) | 26.3 (0.1) | 3.513 | 14.367 | 0.093 | 35 | 116 | 174 | 0.01 |
| ubi.T22Sep/N25D/K29Q ($Tb^{3+}$) | 3.5 (0.1) | 1.3 (0.1) | 5.505 | 1.144 | -8.867 | 150 | 104 | 9 | 0.03 |


[a] The $\Delta\chi$-tensor fits used PCSs measured with $Tb^{3+}$ and $Tm^{3+}$, using $Y^{3+}$ as the diamagnetic reference. The metal coordinates and tensor parameters for the ubiquitin and GB1 mutants are reported relative to the NMR ensemble structure of ubiquitin (PDB ID: 2KOX; Fenwick et al., 2011) and the crystal structure of GB1 (PDB ID: 1PGA; Gallagher et al., 1994), respectively.
[b] Uncertainties (in brackets) were determined from fits obtained by randomly omitting 10 % of the PCS data.
[c] The quality factor was calculated as the root-mean-square deviation between experimental and back-calculated PCSs divided by the root-mean-square of the experimental PCSs.



(a)

(b)

**Figure 2.** Correlation between back-calculated and experimental PCSs, and lanthanide locations on the ubiquitin mutants (a)
E18Sep and (b) E16Q/E18Sep. Left panel: PCS data obtained with $Tb^{3+}$ and $Tm^{3+}$ plotted in red and blue, respectively. Right
panel: Blue and red isosurfaces indicating PCSs of +/-1 ppm, respectively, obtained with $Tb^{3+}$. The side chains of E16 and the
phosphoserine residue in position 18 are shown in a stick representation.





### 3.3 Phosphoserine and aspartate for introducing a lanthanide binding site into GB1

The scheme of combining a phosphoserine with an aspartate helper residue to create a lanthanide binding site was also
successful with the GB1 mutant K10D/T11Sep, where $Tb^{3+}$ ions generated PCSs as large as 0.55 ppm (Fig. 3a, Table S2) and,
as for the ubiquitin mutants, the lanthanide complex was in slow exchange with the free protein. The $\Delta\chi$-tensor fit produced
an excellent correlation between back-calculated and experimental PCSs with a $Q$ factor of 0.01, indicating good
immobilization of the lanthanide ion (Fig. 3c, Table S2). The best fit of the $\Delta\chi$ tensor positioned the lanthanide between the
phosphoserine and aspartic acid residues as expected (Fig. 3e).








**Figure 3.** Close agreement between experimental and back-calculated PCSs of amide protons in the protein GB1 obtained with lanthanide binding sites generated with one or two phosphoserine residues. (a) Left panel: Superimposition of $[^{15}N, ^{1}H]$-HSQC spectra of 0.3 mM solutions of GB1 K10D/T11Sep. The spectra were recorded in the presence of $Tb^{3+}$ (red) or $Y^{3+}$ (black). Lines connect cross-peaks belonging to the same residue in the paramagnetic and diamagnetic samples. Right panel:
Correlation between back-calculated and experimental PCSs and lanthanide locations on the GB1 mutant K10D/T11Sep in complex with $Tb^{3+}$, and PCS isosurfaces plotted on the structure of GB1. Blue and red isosurfaces indicate PCSs of +/-1 ppm, respectively. (b) Same as (a), but for the GB1 K10Sep/T11Sep mutant.

### 3.4 Double-phosphoserine motifs in GB1

Next we assessed the possibility to generate a lanthanide-binding motif by the introduction of two phosphoserine residues. For
comparison with the GB1 mutant K10D/T11Sep, a double-amber mutant of GB1 was made to replace both K10 and T11 by phosphoserine. The protein was obtained in good yield (1.5 mg from 1 litre of cell culture) despite the presence of two amber stop codons. Successful double amber suppression was confirmed by mass spectrometry (Fig. S2a). Following titration with $Tb^{3+}$ ions, we observed PCSs up to 1 ppm (Fig. 3a, Table S2). The $\Delta\chi$-tensor fit indicated that the lanthanide ion binds between the Sep residues in positions 10 and 11 as expected and the agreement between back-calculated and experimental PCSs was
excellent (Fig. 3d and f). The very low $Q$ factor associated with the $\Delta\chi$-tensor fit (Table 2) demonstrates that the PCSs are adequately explained by a single $\Delta\chi$ tensor, indicting the absence of averaging between different tensors arising from translational movements of the paramagnetic centre.

In previous work, we reported that two nitrilotriacetic acid (NTA) tags attached to cysteine residues in positions $i$ and $i+4$ of an $\alpha$-helix yielded larger PCSs with lanthanides than a single NTA tag combined with an acidic helper residue
(Swarbrick et al., 2011). In view of this result, we also attempted to position two phosphoserine residues in positions $i$ and $i+4$ of the $\alpha$-helix of GB1. About 1 mg of GB1 A24Sep/K28Sep was obtained from 300 mL cell culture, and the successful and complete incorporation of two Sep residues was confirmed by mass spectrometry (Fig. S2b).

Following titration with $Tb^{3+}$ ions, PCSs up to 3 ppm were observed (Table S2). Figure 4a shows the PCSs observed with $Tb^{3+}$ and $Tm^{3+}$ ions following titration to a 1:1 lanthanide:protein ratio. Excess lanthanide ion resulted in significant peak
broadening, indicating weak binding of the excess lanthanide ions to less specific sites. The $\Delta\chi$-tensor fits to the crystal structure of GB1 revealed relatively large $\Delta\chi$ tensors and a small $Q$ factor (Fig. 4b and Table 2), indicating good immobilization of the lanthanide ion. The paramagnetic centre identified by the fits placed the lanthanide ions between the two Sep residues as expected (Fig. 4c).






**Table 2.** $\Delta\chi$-tensor parameters of the GB1 mutants K10Sep/T11Sep and A24Sep/K28Sep.[a]

| Mutant | $\Delta\chi_{ax}$ $(10^{-32} \text{ m}^3)$ | $\Delta\chi_{rh}$ $(10^{-32} \text{ m}^3)$ | x (Å) | y (Å) | z (Å) | $\alpha$ (°) | $\beta$ (°) | $\gamma$ (°) | $Q$ |
|---|---|---|---|---|---|---|---|---|---|
| K10Sep/T11Sep ($Tb^{3+}$) | -14.5 (0.1) | -3.2 (0.1) | 27.455 | 13.449 | 12.675 | 88.2 | 12.7 | 154.7 | 0.01 |
| A24Sep/K28Sep ($Tb^{3+}$) | 34.7 (0.6) | 5.3 (0.1) | 17.628 | 34.049 | 21.869 | 178.3 | 46.4 | 69.2 | 0.02 |
| A24Sep/K28Sep ($Tm^{3+}$) | 15.5 (0.4) | 2.5 (0.1) | 17.666 | 34.141 | 21.937 | 178.3 | 46.4 | 46.6 | 0.03 |

[a] The $\Delta\chi$-tensor fits used the crystal structure 1PGA (Gallagher et al., 1994) and the PCSs measured with $Tb^{3+}$ (or $Tm^{3+}$) and
$Y^{3+}$. The quality factor is calculated as the ratio of the root-mean-square deviation between experimental and back-calculated
PCSs and the root-mean-square of the experimental PCSs.



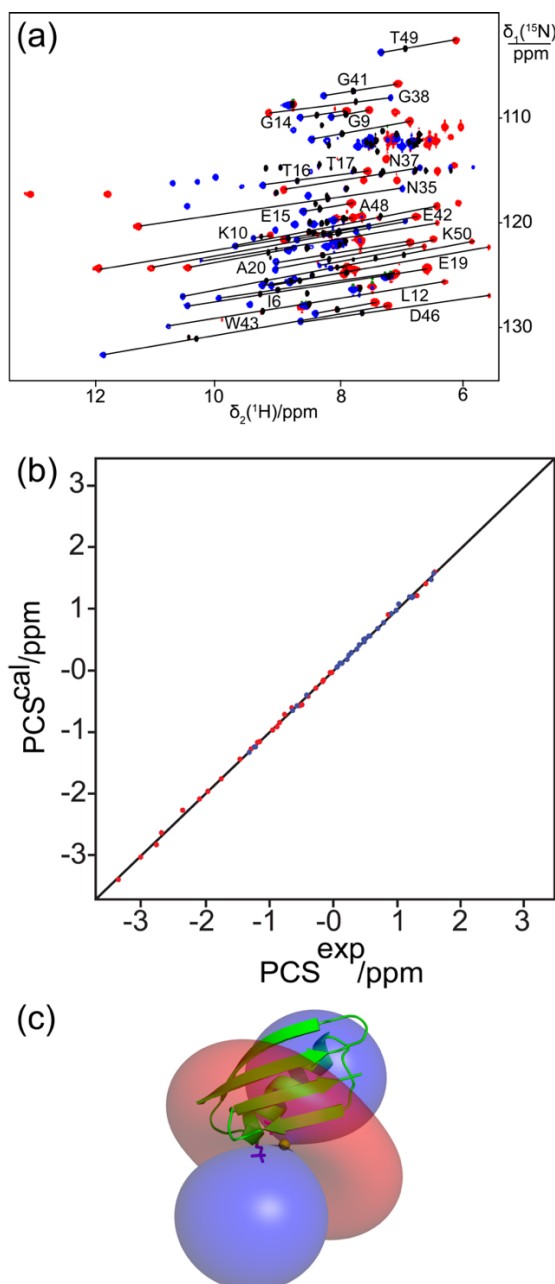

**Figure 4.** The double-phosphoserine mutant GB1 A24Sep/K28Sep generates high-quality PCSs. (a) Superimposition of
[$^{15}$N,$^1$H]-HSQC spectra of 0.3 mM solutions of GB1 A24Sep/K28Sep in the presence of one equivalent of Tb$^{3+}$ (red cross-peaks), Tm$^{3+}$(blue cross-peaks)  or Y$^{3+}$ (black cross-peaks). Lines were drawn to connect selected corresponding cross-peaks observed with diamagnetic and paramagnetic metal ions. (b) Correlation between back-calculated and experimental PCSs. (c) Blue and red isosurfaces indicate PCSs of +/-1 ppm, respectively. The side chains of Sep residues modelled at positions 24 and 28 are highlighted by a stick representation.



### 3.6 Double-phosphoserine incorporation into other proteins

To test the broader validity of double-phosphoserine motifs as lanthanide binding sites, we generated double-amber mutants for double-phosphoserine incorporation in 16 different sites in four different proteins (Fig. S3). The double amber mutations were designed to position two phosphoserine residues in α-helices (positions *i* and *i*+4), loops (positions *i* and *i*+2) and β-strands (positions *i* and *i*+2, as well as two positions located in parallel β-strands). Among the constructs made of GB1, ubiquitin, *E. coli* PpiB, Zika virus NS2B-NS3 protease and the N-terminal ATP-binding domain of *Plasmodium falciparum* Hsp90 (Hsp90-N), *in vivo* expression attempts produced protein only for two of the constructs, namely Hsp90-N S36Sep/D40Sep (where the phosphoserine residues are in an α-helix) and ubiquitin T66/H68 (where the phosphoserine residues are in a β-strand). All the other constructs failed to produce protein. Disappointingly, neither Hsp90-N S36Sep/D40Sep nor ubiquitin T66Sep/H68Sep displayed any PCSs upon titration with paramagnetic lanthanides.

The difficulties to express most of the double-phosphoserine mutants was not due to expression into insoluble inclusion bodies, as we did not find the proteins in the insoluble fraction after cell lysis. As the read-through efficiency of amber stop codons has been reported to depend on neighbouring nucleotides (Pott et al., 2014), we tested the incorporation of Boc-lysine (BoK) to produce ubiquitin A28BoK/D32BoK, *E. coli* PpiB K25BoK/D29BoK and GB1 T51BoK/T53BoK, using a previously published pyrrolysyl-tRNA synthetase/tRNA pair (Bryson et al., 2017). All these proteins were expressed successfully (Fig. S4), demonstrating that the difficulty to express these mutants with two phosphoserine residues arises not simply from the difficulty to read through two amber stop codons in the same gene. These observations suggest that too many negatively charged amino acids located in close proximity interfere with protein folding, making the protein prone to proteolytic degradation during overexpression. Likewise, the ubiquitin mutant A28Sep/D32Sep could not be overexpressed, whereas the single mutant A28Sep was produced in good yield. Unfortunately, ubiquitin A28Sep did not display PCSs following titration with TbCl₃ (data not shown).

### 3.7 Lanthanide binding by three amino acid side-chains

The high failure rate of double-phosphoserine incorporation prompted us to carefully assess the two GB1 double-Sep mutants that did express and deliver PCSs. Notably, both constructs feature an additional glutamate residue near the lanthanide binding site, which could potentially assist with the binding of the lanthanide. Specifically, Glu26 is near the lanthanide binding site of GB1 A24Sep/K28Sep (Fig. 5a), and the side chain of Glu56 is near the loop region harbouring the K10Sep/T11Sep mutations and could point towards the two phosphoserines in the loop (Fig. 5b). Indeed, the lanthanide positions determined by the Δχ-tensor fits are not simply between the two phosphoserine side chains, but are also within reach of the side-chain carboxyl groups of the nearby glutamate residues. The excellent *Q* factors associated with the Δχ-tensor fits (Table 2) suggest that the metal positions are reliable. Notably, none of the other double-phosphoserine mutants investigated (Fig. S3) provided the possibility of additional lanthanide coordination by a negatively charged helper residue. To test the functional importance



of E26, we produced the GB1 A24Sep/K28Sep/E26N triple mutant and probed for lanthanide binding. Indeed, this mutant produced no PCSs upon titration with TbCl₃.

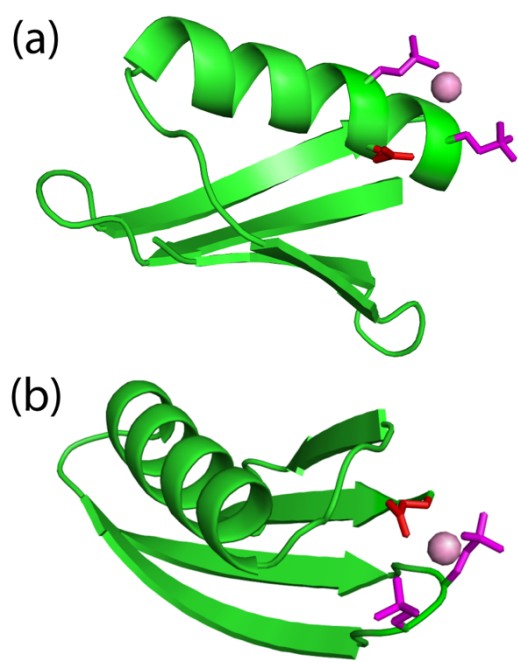

**Figure 5.** An additional glutamate residue acts as a helper residue to bind a lanthanide ion in double-phosphoserine mutants of GB1. (a) GB1 A24Sep/K28Sep with the side chains of the phosphoserine residues (purple) and Glu26 (red) modelled to indicate their possible proximity to a lanthanide ion (purple ball). (b) GB1 K10Sep/T11Sep showing the phosphoserine side chains in purple and E56 in red.

### 3.8 Effect of salt bridges

In wild-type proteins, most aspartate and glutamate residues are located sufficiently close to positively charged side chains that they can engage in salt-bridges. This raises the question, whether such salt bridges can affect the lanthanide binding affinity of sites constructed with negatively charged residues by compensating some of the negative charge. For example, the ubiquitin mutant T22Sep/N25D features a lysine residue (K29) in the α-helix harbouring D25, with the potential to form a salt-bridge (Fig. 6a). To test the effect of this interaction, we replaced K29 by glutamine in the mutant T22Sep/N25D/K29Q. Indeed,

while the mutant T22Sep/N25D displayed only very small PCSs with $Tb^{3+}$ ions if any (Fig. 7a), the mutant T22Sep/N25D/K29Q displayed PCSs up to 0.3 ppm (Fig. 7b, Table S1). Using the NMR ensemble structure of ubiquitin (PDB ID: 2KOX) and the measured PCSs, we determined the metal position in the triple mutant by fitting the $\Delta\chi$ tensor. The correlation between back-calculated and experimental PCSs was excellent, resulting in a $Q$ factor of 0.03 (Fig. 7c, Table 1).





Similarly, fitting of a Δχ tensor to the small PCSs observed for the ubiquitin mutant Q2D/E64Sep (Fig. S5), which

has a lysine residue in position 63, suggested metal coordinates far from the protein, which is a hallmark of a variable metal position (Shishmarev and Otting, 2013). Unfortunately, the attempt to remove the potential salt bridge between K63 and the Sep residue in position 64 in the triple mutant Q2D/K63Q/E64Sep resulted in a construct that failed to express.

Attempts to express the ubiquitin mutant R54Sep and the GB1 mutant K50Sep failed. We speculate that this may be due to the destabilizing effect associated with the disruption of salt bridges involving these sites (Fig. S6a). Conversely, the

ubiquitin mutant T55Sep and the GB1 mutant A24Sep expressed in high yield, but did not display PCSs upon titration with paramagnetic lanthanides. The structure of ubiquitin indicates that a Sep residue in position 55 could form a salt bridge with R54 and the structure of GB1 suggests that a Sep residue in position 24 could form a salt bridge with K28 (Fig. S6b). These results suggest that the expression even of highly stable proteins like ubiquitin and GB1 can be affected by the presence of too many charges in close proximity, while compensating the negative charge density by salt bridges affects lanthanide binding.


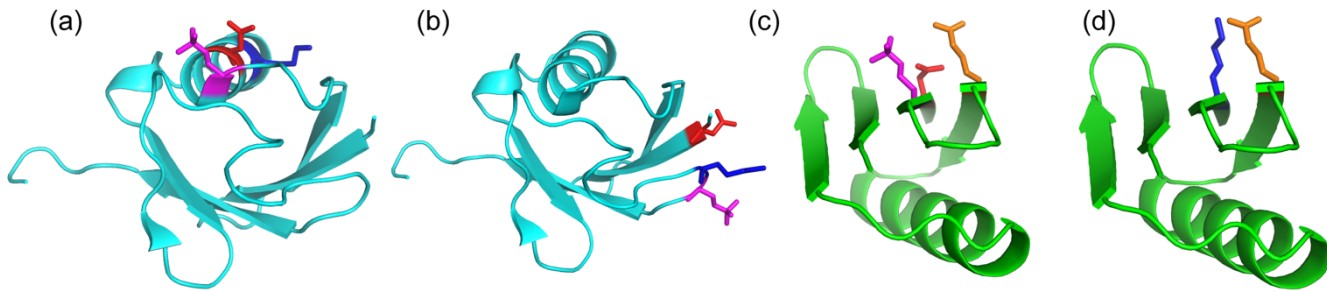

**Figure 6:** Single-site phosphoserine mutants tested for the effect of positively charged residues nearby. The locations of

selected residues are highlighted by displaying side-chain atoms, with phosphoserine in magenta, aspartate in red, glutamate in orange and lysine residues in blue. Side-chain conformations are those of the crystal structure, except for phosphoserine, which was modelled. (a) Ubiquitin T22Sep/N25D. A lysine residue (K29) is located next to the engineered lanthanide binding site in the same α-helix, where it can form a salt-bridge with residue 25. (b) Ubiquitin Q2D/E64Sep. There is a lysine residue in position *i*-1 of the Sep residue. (c) Wild-type GB1 showing the salt bridge between K4 and E15. (d) GB1 K4D/I6Sep.

Introduction of the aspartate and Sep residue resulted in denaturation of the protein (Fig. S7).





**Figure 7.** Breaking a salt bridge in ubiquitin T22Sep/N25D generates a specific lanthanide binding site. (a) Superimposition of [$^{15}$N,$^1$H]-HSQC spectra of 0.3 mM solutions of ubiquitin T22Sep/N25D recorded in the presence of Tb$^{3+}$ (red), or Y$^{3+}$ (black). (b) Same as (a), but for ubiquitin T22Sep/N25D/K29Q. Lines connect cross-peaks belonging to the same residue in the paramagnetic and diamagnetic samples. (c) Correlation between back-calculated and experimental PCSs, (d) Blue and red PCS isosurfaces indicating PCSs of +/-1 ppm, respectively. The side chains of D25 and the phosphoserine residue are highlighted by a stick representation.



### 3.9 Protein unfolding due to charge repulsion

Our failure to produce most of the proteins designed with two phosphoserine residues in close proximity led us to hypothesise that low expression yields could in part be caused by unfolding due to electrostatic repulsion, which would increase

susceptibility to proteolytic degradation during expression in *E. coli*. Supporting evidence came from two observations. First, the GB1 mutant K4D/I6Sep displayed an NMR spectrum characteristic of an unfolded protein (Fig. S7). In wild-type GB1, E15 is in close proximity of K4 (Fig. 6c). By disrupting this salt bridge, the mutant K4D/I6Sep contains several uncompensated negative charges in close proximity (Fig. 6d). Alternatively, E15 could also form a salt bridge with K13. Therefore we attempted to reduce the number of negative charges by producing the mutant K4D/I6Sep/E15Q. Unfortunately, this mutant

failed to express.

       The second piece of evidence for charge-driven unfolding came from phosphoserine mutants of Hsp90-N. Although the wild-type protein can be produced in good yield, the single-phosphoserine mutants K70Q/T71Sep, K70Q/N72Sep, K70Q/N72Sep/D69, N72Sep, Q54D/S57Sep and R98D/S99Sep (Fig. S8a) failed to express, and the mutants D88Sep/N91D, E162D/T163Sep and K160Q/E162D/T163Sep (Fig. S8b) were produced only in very low yields. Only the mutant N91Sep

(Fig. S8c) expressed in sufficient yield for isotope labelling. Its [$^{15}$N,$^{1}$H]-HSQC spectrum showed evidence of partial unfolding, as the signals of many amides vanished while new peaks appeared at chemical shifts characteristic of unfolded proteins. Assignment of the well-resolved cross-peaks by comparison with the wild-type protein showed that the β-sheet of Hsp90-N was conserved in the N91Sep mutant, whereas no evidence was found for structural conservation of the protein region near residue 91 (Fig. S9). Notably, Hsp90-N is a protein of limited stability that is prone to precipitation and degradation

within a couple of days.

### 4 Discussion

The present study shows the potential of phosphoserine for generating lanthanide binding sites on proteins. Using phosphoserine to construct lanthanide binding sites in proteins is uniquely attractive for multiple reasons. (i) Systems are available to genetically encode phosphoserine as an unnatural amino acid for site-specific insertion into polypeptide chains

(Pirman et al., 2015). This provides facile access to the requisite protein mutants. The main alternative way, in which lanthanide ions can be attached to an unnatural amino acid, relies on copper-catalysed click chemistry of alkyne tags with a site-specifically introduced *p*-azidophenylalanine residue (Loh et al., 2013; Loh et al., 2015). In our hands, about half of the proteins have proven to precipitate quantitatively when exposed to the copper catalyst. (ii) Phosphoserine allows to construct the lanthanide binding site without the need of posttranslational modification by a lanthanide-binding chemical tag. Without the

need for chemical modification, the approach is independent of the presence or absence of cysteine residues, or whether the





target protein tolerates the chemicals needed for specific tagging. (iii) The side chain of phosphoserine is relatively short, leading to a lanthanide position close to the protein backbone. This makes it easier to predict the position of the lanthanide ion relative to the protein. While a single phosphoserine residue is not sufficient to bind a lanthanide ion with high affinity, this study shows that a nearby aspartate residue can assist to form a good lanthanide binding site, with the lanthanide ion

coordinated both by the phosphoserine and aspartate residues. This delivers a better localization of the lanthanide ion than most of the chemical tags designed for binding to cysteine residues and, hence, $\Delta\chi$-tensor fits with very small $Q$ factors can be obtained. The small size of the $Q$ factors also indicates that the introduction of a phosphoserine residue does not induce any significant conformational changes in the target protein. High-quality $\Delta\chi$-tensor fits open the door for exploiting PCSs as accurate long-range restraints in structural biology.

Exceptionally low $Q$ factors were obtained for a lanthanide binding site in GB1, which was made of two phosphoserine residues in positions $i$ and $i+4$ of the α-helix together with Glu15. The site also generated relatively large $\Delta\chi$ tensors, indicating excellent immobilization of the metal ion relative to the protein (Shishmarev and Otting, 2013) as well as full conservation of the 3D structure of the protein. Two phosphoserine residues in a loop region of GB1 also produced a very small $Q$ factor. It was disappointing, however, that attempts to produce other proteins with two phosphoserine residues met

with a high failure rate. This may be explained by a failure to fold due too many negatively charged residues located in close proximity (Baneyx and Mujacic, 2004), resulting in degradation of the proteins during expression.

We succeeded to produce double-phosphoserine mutants of only two proteins other than GB1. These were the Hsp90-N mutant S36Sep/D40Sep and the ubiquitin mutant T66Sep/H68Sep. Both expressed in good yield but failed to produce PCSs with lanthanides. Furthermore, the absence of paramagnetic relaxation enhancements upon titration with lanthanides indicated

the failure to bind. Inspection of the 3D structures of these proteins indicated that nearby residues with positively charged side chains were in positions capable of at least partially compensating the negative charges of the phosphoserine residues. The fact that the ubiquitin mutant T22Sep/N25D/K29Q produced much better PCSs than the mutant T22Sep/N25D (Fig. 7) illustrates the potentially detrimental effect of salt-bridges on lanthanide binding.

In summary, when designing lanthanide binding sites with phosphoserine residues, a single phosphoserine residue in

combination with an aspartate can deliver binding affinities in the micromolar range, but positively charged side chains near the designed lanthanide binding site can compromise its ability to bind lanthanides. At the same time, the difficulty to produce proteins that contain many negatively charged residues in close proximity points to the importance of salt bridges to ensure the structural integrity of proteins.

## 5 Conclusions

The present study demonstrates, for the first time, that a lanthanide binding motif can be introduced into a protein via genetically encoded unnatural amino acids without further chemical modification. It is particularly promising that the lanthanide binding motif can be generated in either an α-helix or a loop region by a single phosphoserine residue combined

with an aspartate, provided these residues are not engaged in salt-bridges. While two phosphoserine residues potentially bind lanthanide ions even more strongly, too many negatively charged residues in close proximity tend to severely affect the *in vivo*

expression yields as well as the folding of the target protein. For proteins, where lanthanide binding sites can successfully be installed with the help of phosphoserine residues, however, $\Delta\chi$ tensors of extraordinary quality can be obtained.

**Supplement.** The supplement related to this article is available online at:…

**Code/Data availability.** The NMR spectra are available at 10.6084/m9.figshare.13159748.

**Author contributions.** GO initiated the project and edited the final version of the manuscript. SMT performed most experiments and wrote the first version of the manuscript. MM produced and analysed all protein mutants with two Sep residues. CTL established the phosphoserine incorporation protocols and performed the first successful PCS measurements

with lanthanide ions. IA produced wild-type and mutant samples of Hsp90-N, provided NMR resonance assignments and analysed partially unfolded mutants.

**Competing interests.** None.

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
