# Peer review of "Phosphoserine for the generation of lanthanide binding sites on proteins for paramagnetic NMR"

_Magnetic Resonance, 2020_

## Referee Comment (RC1) · Marcellus Ubbink (Referee) · 17 Nov 2020

This manuscript describes a new way of tagging proteins with lanthanoids. It is an original new approach yielding excellent results on some systems. It is highly appreciated that the authors also discuss the systems for which the method does not work, describing its limitations as well as its advantages. The methods are described in sufficient detail and all results appear sound. Thus, the work is a useful addition to the field of Magnetic Resonance.

There are only a few comments that the authors may address for further improvement:

[Figure]

l. 100: A 100-fold excess of TEV was used, that seems an awful lot for an enzyme. How was it removed? Was second NTA column used?

l. 108, 109: The Bruker line of consoles is called Avance, not Advance

Table 1: It would be useful to add the number of PCS used in each calculation in an extra column. Also, the tensors for Tm3+ are very low indeed compared to those for Tb3+. Other tags, such as CLaNP give very high values for Tm3+ ($\sim$ 55 x 10ˆ-32). Do the authors know why these differ so much? The Q-values are very low, as mentioned, and all but one are 0.03, yet looking at the plot for Ubi E16Q/E18Sep(Tm3+) the spread looks clearly larger than for others (Fig. 2). How can that be?

Fig. S1: Can you indicate the fitted parameters for the ITC? Were n, Delta-H and K(D) all fitted? What are the results?

[Figure]

---

## Referee Comment (RC2) · Claudio Luchinat (Referee) · 25 Nov 2020

A new lanthanoid binding tag is proposed, which does not require cysteine residues for attachment to the protein, based on the incorporation of phosphoserine in close proximity of negatively charged residues. The vicinity of the binding site to the protein backbone is expected to ensure a good immobilization of the tag. On the other hand, the authors show that a double phosphoserine mutation can increase the magnitude of the magnetic susceptibility anisotropy, and thus of the pcs, but also show that it is difficult to produce most of these mutants.

The tag here presented is of potential importance for the improvement of tagging strategies of proteins; the manuscript is interesting and well written, and thus recommended for publication.

The authors should comment that the axial anisotropies of the proposed tag attached to ubiquitin with a single phosphoserine mutation are significantly smaller than those of other previously proposed rigid tags (more than a factor 2 for Tb probes, more than a factor 5 for Tm), and should discuss the origin of this difference.

The tensor for the GB1 K10D/T11Sep(Tb3+) should be reported with an axial component of -33.7 and a rhombic component of 14.7 to fulfill the axis labeling convention providing a rhombic component up to 2/3 of the axial component in absolute value. If the authors prefer to report the tensor as in Table 1, they should at least explain why. In any case, the tensor anisotropy is surprisingly large considering that the measured pcs span a range smaller than that measured for ubiquitin, and surprisingly rather rhombic. In the double phosphoserine K10Sep/T11Sep(Tb3+) mutant, the measured values of the pcs span a range which is roughly double, but the tensor is less than half with respect to that of K10D/T11Sep(Tb3+). Please, double check that no mix-up of data has occurred.

Can you comment on the reason of the different sign of the tensor axial components between K10Sep/T11Sep(Tb3+) and A24Sep/K28Sep(Tb3+)? On the other hand, the sign of the axial components of Tb and Tm are usually opposite. Why are they the same in A24Sep/K28Sep?

Minor points:

Pag.2, line 1: "As lanthanide ions display particularly large..." not all lanthanoids, only some of them!

Pag. 2, line 2: "While paramagnetic lanthanide ions generate paramagnetic relaxation enhancements (PRE) in the protein irrespective of metal mobility" This sentence may be read that PREs do not depend on mobility, which is slightly inaccurate, because

internal mobility changes the correlation time of dipole-dipole relaxation (see Fragai et al. Coord. Chem. Rev. 2013, 257, 2652 for a thorough discussion). Please, clarify this point.

Caption to Fig. 3: please indicate all panel letters.

───────────────────────────

---

## Referee Comment (RC3) · Marcellus Ubbink (Referee) · 30 Nov 2020

I thank the authors for the replies to my comments. They are clear and satisfactory. I have no more comments. Nice work!

---

## Referee Comment (RC4) · Claudio Luchinat (Referee) · 30 Nov 2020

The authors addressed all my comments satisfactorily. I recommend publication of this very nice work.

---

## Author Comment (AC1) · 30 Nov 2020

We are grateful for the insightful comments and identifying errors.

Line 100: A 100-fold excess of TEV was used, that seems an awful lot for an enzyme. How was it removed? Was second NTA column used?

Response: We appreciate detection of this error! TEV protease was added in 0.1 molar ratio to remove the His6 tag. In addition, a second 5 mL Ni-NTA column was used to remove the TEV protease from the protein. We propose to write in the revised manuscript: "the protein was dialysed into TEV protease buffer (50 mM Tris-HCl, pH

8.0, 300 mM NaCl and 1 mM beta-mercaptoethanol) to remove the His6-tag by digestion with TEV protease overnight at 4 °C. His6-tagged TEV protease was added in 0.1 molar ratio. The protease and cleaved His6-tag were removed by running the sample again over a Ni-NTA column."

Lines 108, 109: The Bruker line of consoles is called Avance, not Advance

Response: Thank you for pointing out this typo, which will be corrected in the revised manuscript.

Table 1: It would be useful to add the number of PCS used in each calculation in an extra column. Also, the tensors for Tm3+ are very low indeed compared to those for Tb3+. Other tags, such as CLaNP give very high values for Tm3+ (-55 x 10ˆ-32). Do the authors know why these differ so much? The Q-values are very low, as mentioned, and all but one are 0.03, yet looking at the plot for Ubi E16Q/E18Sep(Tm3+) the spread looks clearly larger than for others (Fig. 2). How can that be?

Response: In the revised manuscript, we will replace Tables 1 and 2 in the main text as shown in the figures attached, with changes highlighted in yellow. All PCSs used for the tensor fits of the various ubiquitin and GB1 mutants are already listed in Tables S1 and S2, respectively.

Indeed, the difference in DeltaChi tensors obtained with Tm3+ and Tb3+ was larger than expected for the single-Sep mutants (but not for the GB1 mutant A24Sep/K28Sep). We observed previously that the ratio between the axial tensor components of these two ions can vary between different tags and even for the same tag at different sites of a protein (C.-T. Loh, B. Graham, E. H. Abdelkader, K. L. Tuck, G. Otting (2015) Generation of pseudocontact shifts in proteins with lanthanides using small "clickable" nitrilotriacetic acid and iminodiacetic acid tags Chem. Eur. J. 21, 5084-5092). These differences are not an artifact of fitting the tensors for Tm3+ and Tb3+ independently, as the fits yielded very similar coordinates for both metal ions. We do not understand the origin of these effects. It would help, if the effect of the ligand field

could be predicted by quantum-mechanical calculations, but we were told by experts in the field that this is prohibitively difficult for lanthanide ions.

In the revised version, we propose to add the following paragraph in line 371: "The DeltaChi tensors obtained with Tm3+ instead of Tb3+ ions were unexpectedly low for the single-Sep mutants, but not for the GB1 mutant A24Sep/K28Sep. We observed previously that the ratio between the DeltaChi_axial components of these two ions can vary between different tags and even for the same tag at different sites of a protein (Loh et al., 2015). These differences are not an artifact of fitting the tensors for Tm3+ and Tb3+ independently, as, with the exception of ubiquitin E18Sep, the fits converged to very similar metal positions (Tables 1 and 2). We do not understand the origin of different magnitudes of Chi-tensor anisotropies for Tm3+ and Tb3+ ions. In addition, much larger DeltaChi tensors have been reported for sterically rigid cyclen tags (Joss and Häussinger, 2019), suggesting that a rigid ligand field promotes large DeltaChi tensors."

The quality factor for the DeltaChi-tensor fit of Tb3+ in ubiquitin E18Sep differed from that of ubiquitin E16Q/E18Sep in the second digit, which was not displayed. On re-inspection, we noted that the Q factors had been rounded incorrectly: the Q factor for worse-fitting data should have been reported as 0.04 instead of 0.03. This was fixed in Table 1 attached and will be fixed in the revised version of the manuscript. In the case of Tm3+, the back-calculated and experimental PCSs correlate similarly well in Fig. 2, and the Q factors were correspondingly similar.
* * *
170 **Table 1.** Δχ-tensor parameters of the ubiquitin mutants E18Sep, E16Q/E18Sep and T22Sep/N25D/K29Q and the GB1 mutant K10D/T11Sep complexed with $Tb^{3+}$ and $Tm^{3+}$ ions.[a]

| Protein | N[b] | $\Delta\chi_{ax}$[c] ($10^{-32}$ m³) | $\Delta\chi_{rh}$[c] ($10^{-32}$ m³) | x (Å) | y (Å) | z (Å) | α (°) | β (°) | γ (°) | Q[d] |
|---|---|---|---|---|---|---|---|---|---|---|
| ubiquitin E18Sep ($Tb^{3+}$) | 20 | 17.1 (0.6) | 2.8 (0.3) | 10.095 | -1.846 | -11.711 | 170 | 138 | 50 | 0.03 |
| ubiquitin E18Sep ($Tm^{3+}$) | 27 | -2.7 (0.1) | -1.0 (0.1) | 9.463 | -0.674 | -12.207 | 168 | 129 | 49 | 0.03 |
| ubiquitin E16Q/E18Sep ($Tb^{3+}$) | 27 | 15.9 (0.6) | 3.4 (0.8) | 9.695 | -1.754 | -11.833 | 162 | 135 | 37 | 0.04 |
| ubiquitin E16Q/E18Sep ($Tm^{3+}$) | 28 | -4.5 (0.1) | -2.1 (0.1) | 9.441 | -1.902 | -11.918 | 164 | 131 | 59 | 0.03 |
| GB1 K10D/T11Sep ($Tb^{3+}$) | 26 | 7.3 (0.1) | 1.6 (0.1) | 3.513 | 14.367 | 0.093 | 35 | 116 | 174 | 0.01 |
| ubi. T22Sep/N25D/K29Q ($Tb^{3+}$) | 20 | 3.5 (0.1) | 1.3 (0.1) | 5.505 | 1.144 | -8.867 | 150 | 104 | 9 | 0.03 |

[a] The Δχ-tensor fits used PCSs measured with $Tb^{3+}$ and $Tm^{3+}$, using $Y^{3+}$ as the diamagnetic reference. The metal coordinates
175 and tensor parameters for the ubiquitin and GB1 mutants are reported relative to the NMR ensemble structure of ubiquitin
(PDB ID: 2KOX; Fenwick et al., 2011) and the crystal structure of GB1 (PDB ID: 1PGA; Gallagher et al., 1994), respectively.
[b] N: number of PCSs used in the fit.
[c] Uncertainties (in brackets) were determined from fits obtained by randomly omitting 10 % of the PCS data.
[d] The quality factor was calculated as the root-mean-square deviation between experimental and back-calculated PCSs divided
180 by the root-mean-square of the experimental PCSs.

**Fig. 1.** revised Table 1

**Table 2.** $\Delta\chi$-tensor parameters of the GB1 mutants K10Sep/T11Sep and A24Sep/K28Sep.[a]

| Mutant | $N$ | $\Delta\chi_{ax}$ (10⁻³² m³) | $\Delta\chi_{rh}$ (10⁻³² m³) | x (Å) | y (Å) | z (Å) | α (°) | β (°) | γ (°) | $Q$ |
|---|---|---|---|---|---|---|---|---|---|---|
| K10Sep/T11Sep (Tb³⁺) | 31 | -14.5 (0.1) | -3.2 (0.1) | 27.455 | 13.449 | 12.675 | 88 | 13 | 155 | 0.01 |
| A24Sep/K28Sep (Tb³⁻) | 34 | 34.7 (0.6) | 5.3 (0.1) | 17.628 | 34.049 | 21.869 | 178 | 46 | 69 | 0.02 |
| A24Sep/K28Sep (Tm³⁺) | 31 | -15.5 (0.4) | -2.5 (0.1) | 17.666 | 34.141 | 21.937 | 178 | 46 | 47 | 0.03 |

230    [a] The $\Delta\chi$-tensor fits used the crystal structure 1PGA (Gallagher et al., 1994) and the PCSs measured with Tb³⁺ (or Tm³⁺) and Y³⁺. See footnotes b-d of Table 1 for further details.

**Fig. 2.** revised Table 2

---

## Author Comment (AC2) · 30 Nov 2020

We are grateful for the insightful comments and identifying errors.

Comment 1: The authors should comment that the axial anisotropies of the proposed tag attached to ubiquitin with a single phosphoserine mutation are significantly smaller than those of other previously proposed rigid tags (more than a factor 2 for Tb probes, more than a factor 5 for Tm), and should discuss the origin of this difference.

Response: The same point was picked up by Marcellus Ubbink and our response is copied here.

Indeed, the difference in DeltaChi tensors obtained with Tm3+ and Tb3+ was larger than expected for the single-Sep mutants (but not for the GB1 mutant A24Sep/K28Sep). We observed previously that the ratio between the axial tensor components of these two ions can vary between different tags and even for the same tag at different sites of a protein (C.-T. Loh, B. Graham, E. H. Abdelkader, K. L. Tuck, G. Otting (2015) Generation of pseudocontact shifts in proteins with lanthanides using small "clickable" nitrilotriacetic acid and iminodiacetic acid tags Chem. Eur. J. 21, 5084-5092). These differences are not an artifact of fitting the tensors for Tm3+ and Tb3+ independently, as the fits yielded very similar coordinates for both metal ions. We do not understand the origin of these effects. It would help, if the effect of the ligand field could be predicted by quantum-mechanical calculations, but we were told by experts in the field that this is prohibitively difficult for lanthanide ions.

In the revised version, we propose to add the following paragraph in line 371: "The DeltaChi tensors obtained with Tm3+ instead of Tb3+ ions were unexpectedly low for the single-Sep mutants, but not for the GB1 mutant A24Sep/K28Sep. We observed previously that the ratio between the DeltaChi_axial components of these two ions can vary between different tags and even for the same tag at different sites of a protein (Loh et al., 2015). These differences are not an artifact of fitting the tensors for Tm3+ and Tb3+ independently, as, with the exception of ubiquitin E18Sep, the fits converged to very similar metal positions (Tables 1 and 2). We do not understand the origin of different magnitudes of Chi-tensor anisotropies for Tm3+ and Tb3+ ions. In addition, much larger DeltaChi tensors have been reported for sterically rigid cyclen tags (Joss and Häussinger, 2019), suggesting that a rigid ligand field promotes large DeltaChi tensors."

Comment 2: The tensor for the GB1 K10D/T11Sep(Tb3+) should be reported with an axial component of -33.7 and a rhombic component of 14.7 to fulfill the axis labeling convention providing a rhombic component up to 2/3 of the axial component in absolute value. If the authors prefer to report the tensor as in Table 1, they should at least

explain why. In any case, the tensor anisotropy is surprisingly large considering that the measured pcs span a range smaller than that measured for ubiquitin, and surprisingly rather rhombic. In the double phosphoserine K10Sep/T11Sep (Tb3+) mutant, the measured values of the pcs span a range which is roughly double, but the tensor is less than half with respect to that of K10D/T11Sep(Tb3+). Please, double check that no mix-up of data has occurred.

Response: Thank you for alerting us to this typo. The correct numbers for DeltaChi_axial and DeltaChi_rhombic are 7.3 and 1.6, respectively.

Comment 3: Can you comment on the reason of the different sign of the tensor axial components between K10Sep/T11Sep(Tb3+) and A24Sep/K28Sep(Tb3+)? On the other hand, the sign of the axial components of Tb and Tm are usually opposite. Why are they the same in A24Sep/K28Sep.

Response: We do not understand the reason for the sign change in the tensor for Tb3+ between the K10Sep/T11Sep and A24Sep/K28Sep mutants. We double-checked and couldn't find an error. The signs were indeed wrong for the Tm3+ tensor associated with GB1 A24Sep/K28Sep(Tm3+): the correct values for the axial and rhombic components are -15.5 and -2.5, respectively.

In the revised version, we will display the isosurfaces also for Tm3+ in Figures 2, 3 and 4 to illustrate the degree of orthogonality of the tensors between Tm3+ and Tb3+ (revised Figures attached).

Comment 4: Minor points: Pag.2, line 1: "As lanthanide ions display particularly large. . ." not all lanthanoids, only some of them! Pag. 2, line 2: "While paramagnetic lanthanide ions generate paramagnetic relaxation enhancements (PRE) in the protein irrespective of metal mobility" This sentence may be read that PREs do not depend on mobility, which is slightly inaccurate, because internal mobility changes the correlation time of dipole-dipole relaxation (see Fragai et al. Coord. Chem. Rev. 2013, 257, 2652 for a thorough discussion). Please, clarify this point. Caption to Fig. 3: please indicate

all panel letters.

Response: In the revised version, we propose the following changes. Page 2, line 1: "As many lanthanide ions display particularly large. . ." Page 2, paragraph 2: "Paramagnetic lanthanide ions always generate paramagnetic relaxation enhancements (PRE) in the protein, which vary relatively little with minor movements of the metal ion. In contrast, PCSs can decrease dramatically if the lanthanide complex reorientates relative to the protein."
* * *
170 **Table 1.** $\Delta\chi$-tensor parameters of the ubiquitin mutants E18Sep, E16Q/E18Sep and T22Sep/N25D/K29Q and the GB1 mutant K10D/T11Sep complexed with $Tb^{3+}$ and $Tm^{3+}$ ions.[a]

| Protein | N[b] | $\Delta\chi_{ax}$[c] ($10^{-32}$ m³) | $\Delta\chi_{rh}$[c] ($10^{-32}$ m³) | x (Å) | y (Å) | z (Å) | α (°) | β (°) | γ (°) | Q[d] |
|---|---|---|---|---|---|---|---|---|---|---|
| ubiquitin E18Sep ($Tb^{3+}$) | 20 | 17.1 (0.6) | 2.8 (0.3) | 10.095 | -1.846 | -11.711 | 170 | 138 | 50 | 0.03 |
| ubiquitin E18Sep ($Tm^{3+}$) | 27 | -2.7 (0.1) | -1.0 (0.1) | 9.463 | -0.674 | -12.207 | 168 | 129 | 49 | 0.03 |
| ubiquitin E16Q/E18Sep ($Tb^{3+}$) | 27 | 15.9 (0.6) | 3.4 (0.8) | 9.695 | -1.754 | -11.833 | 162 | 135 | 37 | 0.04 |
| ubiquitin E16Q/E18Sep ($Tm^{3+}$) | 28 | -4.5 (0.1) | -2.1 (0.1) | 9.441 | -1.902 | -11.918 | 164 | 131 | 59 | 0.03 |
| GB1 K10D/T11Sep ($Tb^{3+}$) | 26 | 7.3 (0.1) | 1.6 (0.1) | 3.513 | 14.367 | 0.093 | 35 | 116 | 174 | 0.01 |
| ubi. T22Sep/N25D/K29Q ($Tb^{3+}$) | 20 | 3.5 (0.1) | 1.3 (0.1) | 5.505 | 1.144 | -8.867 | 150 | 104 | 9 | 0.03 |

[a] The $\Delta\chi$-tensor fits used PCSs measured with $Tb^{3+}$ and $Tm^{3+}$, using $Y^{3+}$ as the diamagnetic reference. The metal coordinates
175 and tensor parameters for the ubiquitin and GB1 mutants are reported relative to the NMR ensemble structure of ubiquitin
(PDB ID: 2KOX; Fenwick et al., 2011) and the crystal structure of GB1 (PDB ID: 1PGA; Gallagher et al., 1994), respectively.
[b] N: number of PCSs used in the fit.
[c] Uncertainties (in brackets) were determined from fits obtained by randomly omitting 10 % of the PCS data.
[d] The quality factor was calculated as the root-mean-square deviation between experimental and back-calculated PCSs divided
180 by the root-mean-square of the experimental PCSs.

**Fig. 1.** revised Table 1

**Table 2.** $\Delta\chi$-tensor parameters of the GB1 mutants K10Sep/T11Sep and A24Sep/K28Sep.[a]

| Mutant | N | $\Delta\chi_{ax}$ $(10^{-32}\ m^3)$ | $\Delta\chi_{rh}$ $(10^{-32}\ m^3)$ | x (Å) | y (Å) | z (Å) | α (°) | β (°) | γ (°) | Q |
|---|---|---|---|---|---|---|---|---|---|---|
| K10Sep/T11Sep (Tb³⁺) | 31 | -14.5 (0.1) | -3.2 (0.1) | 27.455 | 13.449 | 12.675 | 88 | 13 | 155 | 0.01 |
| A24Sep/K28Sep (Tb³⁻) | 34 | 34.7 (0.6) | 5.3 (0.1) | 17.628 | 34.049 | 21.869 | 178 | 46 | 69 | 0.02 |
| A24Sep/K28Sep (Tm³⁺) | 31 | -15.5 (0.4) | -2.5 (0.1) | 17.666 | 34.141 | 21.937 | 178 | 46 | 47 | 0.03 |

230    [a] The $\Delta\chi$-tensor fits used the crystal structure 1PGA (Gallagher et al., 1994) and the PCSs measured with Tb³⁺ (or Tm³⁺) and Y³⁺. See footnotes b-d of Table 1 for further details.

**Fig. 2.** revised Table 2

[Figure]

**Figure 2.** Correlation between back-calculated and experimental PCSs, and lanthanide locations on the ubiquitin mutants (a) E18Sep and (b) E16Q/E18Sep. Left panel: PCS data obtained with Tb$^{3+}$ and Tm$^{3+}$ plotted in red and blue, respectively. Right panel: Blue and red PCS isosurfaces, plotted on the protein structure and indicating PCSs of +/-1 ppm, respectively. The isosurfaces illustrate the $\Delta\chi$ tensors obtained with Tb$^{3+}$ (upper structure) and Tm$^{3+}$ (lower structure). The side chains of E16 and the phosphoserine residue in position 18 are shown in a stick representation.

**Fig. 3.** revised Figure 2

[Figure]

195

11

200

**Figure 3.** Close agreement between experimental and back-calculated PCSs of amide protons in the protein GB1 obtained with lanthanide binding sites generated with one or two phosphoserine residues. (a) and (b) Superimposition of [$^{15}$N,$^{1}$H]-HSQC spectra of 0.3 mM solutions of GB1 K10D/T11Sep and GB1 K10Sep/T11Sep, respectively. The spectra were recorded in the presence of Tb$^{3+}$ (red) or Y$^{3+}$ (black). Lines connect cross-peaks belonging to the same residue in the paramagnetic and diamagnetic samples. (c) and (d) Correlation between back-calculated and experimental PCSs for GB1 K10D/T11Sep and GB1 K10Sep/T11Sep, respectively. (e) and (f) Location of the Tb$^{3+}$ ion on the GB1 mutants K10D/T11Sep and K10Sep/T11Sep, respectively, and PCS isosurfaces plotted on the structure of GB1. Blue and red isosurfaces indicate PCSs of +/-1 ppm, respectively.

**Fig. 4.** revised Figure 3

[Figure]

**Figure 4.** The double-phosphoserine mutant GB1 A24Sep/K28Sep generates high-quality PCSs. (a) Superimposition of
235   [$^{15}$N,$^{1}$H]-HSQC spectra of 0.3 mM solutions of GB1 A24Sep/K28Sep in the presence of one equivalent of Tb$^{3+}$ (red crosspeaks), Tm$^{3+}$(blue cross-peaks) or Y$^{3+}$ (black cross-peaks). Lines were drawn to connect selected corresponding cross-peaks
observed with diamagnetic and paramagnetic metal ions. (b) Correlation between back-calculated and experimental PCSs. (c)
Blue and red isosurfaces indicating PCSs of +/-1 ppm, respectively, as determined by the Δχ-tensors of Tb$^{3+}$ (left) and Tm$^{3+}$
(right). The side chains of Sep residues modelled at positions 24 and 28 are highlighted by a stick representation.

**Fig. 5.** revised Figure 4

---

## Author Comment (AC3) · 30 Nov 2020

The revised Fig. S1 belonging to the previous author comment.

[Figure]

**Figure S1.** Representative isothermal titration calorimetry experiments of ubiquitin E18Sep titrated with LnCl₃. (a) Cell = 150 μM ubiquitin E18Sep; syringe = 2.7 mM TbCl₃. (b) Cell = 150 μM ubiquitin E18Sep; syringe = 2.7 mM TmCl₃. The top panel shows the baseline-corrected power traces. The middle panel displays the heat data and best fit. The bottom panel shows the residual of the fit. Error bars calculated by the program NITPIC (Keller et al., 2015) indicate the standard error in the integration of the peaks. DP denotes the power differential between the reference and sample cells to maintain a zero temperature difference between the cells.

Values for the dissociation constant $K_d$ were derived from global fits to data from two and three different measurements with $Tb^{3+}$ and $Tm^{3+}$, respectively. Fits were performed either with inclusion of the binding stoichiometry $n$ as a fitting parameter or setting $n = 1$, with the result shown underneath.

| Fitted parameters | $Tb^{3+}$ | | $Tm^{3+}$ | |
|---|---|---|---|---|
| | setting $n = 1$ | fitting $n$[a] | setting $n = 1$ | fitting $n$[b] |
| ΔH (kJ mol⁻¹) | 15 | 23 | 20 | 12 |
| ΔS (Jmol⁻¹K) | 137 | 161 | 143 | 128 |
| $K_d$ (μM) | 25 | 42 | 133 | 32 |

[a] The fit yielded $n = 0.7$.
[b] The fit yielded $n = 1.4$.

**Fig. 1.** revised Figure S1

---

## Author Response (AR1)

Below we copied our responses to the reviewers' comments, which we already posted during the discussion phase. This is followed by the revised manuscript and supporting information, with the changes made highlighted in yellow.

Response to the comments made by Marcellus Ubbink:

We are grateful for the insightful comments and identifying errors.

Line 100: A 100-fold excess of TEV was used, that seems an awful lot for an enzyme. How was it removed? Was second NTA column used?

Response: We appreciate detection of this error! TEV protease was added in 0.1 molar ratio to remove the $His_6$ tag. In addition, a second 5 mL Ni-NTA column was used to remove the TEV protease from the protein. We propose to write in the revised manuscript: "the protein was dialysed into TEV protease buffer (50 mM Tris-HCl, pH 8.0, 300 mM NaCl and 1 mM beta-mercaptoethanol) to remove the $His_6$-tag by digestion with TEV protease overnight at 4 °C. $His_6$-tagged TEV protease was added in 0.1 molar ratio. The protease and cleaved $His_6$-tag were removed by running the sample again over a Ni-NTA column."

Lines 108, 109: The Bruker line of consoles is called Avance, not Advance

Response: Thank you for pointing out this typo, which will be corrected in the revised manuscript.

Table 1: It would be useful to add the number of PCS used in each calculation in an extra column. Also, the tensors for Tm3+ are very low indeed compared to those for Tb3+. Other tags, such as CLaNP give very high values for Tm3+ ($\sim$ 55 x 10ˆ-32). Do the authors know why these differ so much? The Q-values are very low, as mentioned, and all but one are 0.03, yet looking at the plot for Ubi E16Q/E18Sep($Tm^{3+}$) the spread looks clearly larger than for others (Fig. 2). How can that be?

Response: In the revised manuscript, we will replace Tables 1 and 2 in the main text as shown in the figures attached, with changes highlighted in yellow. All PCSs used for the tensor fits of the various ubiquitin and GB1 mutants are already listed in Tables S1 and S2, respectively.

Indeed, the difference in DeltaChi tensors obtained with $Tm^{3+}$ and $Tb^{3+}$ was larger than expected for the single-Sep mutants (but not for the GB1 mutant A24Sep/K28Sep). We observed previously that the ratio between the axial tensor components of these two ions can vary between different tags and even for the same tag at different sites of a protein (C.-T. Loh, B. Graham, E. H. Abdelkader, K. L. Tuck, G. Otting (2015) Generation of pseudocontact shifts in proteins with lanthanides using small "clickable" nitrilotriacetic acid and iminodiacetic acid tags *Chem. Eur. J.* 21, 5084-5092). These differences are not an artifact of fitting the tensors for $Tm^{3+}$ and $Tb^{3+}$ independently, as the fits yielded very similar coordinates for both metal ions. We do not understand the origin of these effects. It would help, if the effect of the ligand field could be predicted by quantum-mechanical calculations, but we were told by experts in the field that this is prohibitively difficult for lanthanide ions.

In the revised version, we propose to add the following paragraph in line 371:

"The DeltaChi tensors obtained with $Tm^{3+}$ instead of $Tb^{3+}$ ions were unexpectedly low for the single-Sep mutants, but not for the GB1 mutant A24Sep/K28Sep. We observed previously that the ratio between the DeltaChi_axial components of these two ions can vary between different tags and even for the same tag at different sites of a protein (Loh et al., 2015). These differences are not an artifact of fitting the tensors for $Tm^{3+}$ and $Tb^{3+}$ independently, as, with the exception of ubiquitin E18Sep, the fits converged to very similar metal positions (Tables 1 and 2). We do not understand the origin of different magnitudes of Chi-tensor anisotropies for $Tm^{3+}$ and $Tb^{3+}$ ions. In addition, much larger DeltaChi tensors have been reported for sterically rigid cyclen tags (Joss and Häussinger, 2019), suggesting that a rigid ligand field promotes large DeltaChi tensors."

The quality factor for the DeltaChi-tensor fit of $Tb^{3+}$ in ubiquitin E18Sep differed from that of ubiquitin E16Q/E18Sep in the second digit, which was not displayed. On re-inspection, we noted that the $Q$ factors had been rounded incorrectly: the $Q$ factor for worse-fitting data should have been reported as 0.04 instead of 0.03. This was fixed in Table 1 attached and will be fixed in the revised version of the manuscript. In the case of $Tm^{3+}$, the back-calculated and experimental PCSs correlate similarly well in Fig. 2, and the $Q$ factors were correspondingly similar.

Fig. S1: Can you indicate the fitted parameters for the ITC? Were n, Delta-H and K(D) all fitted? What are the results?

Response: The ITC measurements proved to be difficult. The data below are from three different measurements with $Tb^{3+}$ and two different measurements with $Tm^{3+}$. The $K_d$ values were derived from global fits. Fits were performed either with inclusion of the binding stoichiometry $n$ as a fitting parameter or setting $n = 1$. Both results are included in the table attached. It is clear that these data do not yield the dissociation constant $K_d$ with the accuracy suggested by the original manuscript. Therefore, we will change the sentence in line 145 to "Isothermal calorimetric experiments with $Tb^{3+}$ and $Tm^{3+}$ ions indicated dissociation constants of about 30–50 micromolar (Fig. S1)." and change the legend of Fig. S1 as shown in the attachment.

Response to the comments made by Claudio Luchinat:

We are grateful for the insightful comments and identifying errors.

Comment 1: The authors should comment that the axial anisotropies of the proposed tag attached to ubiquitin with a single phosphoserine mutation are significantly smaller than those of other previously proposed rigid tags (more than a factor 2 for Tb probes, more than a factor 5 for Tm), and should discuss the origin of this difference.

Response: The same point was picked up by Marcellus Ubbink and our response is copied here.

Indeed, the difference in DeltaChi tensors obtained with $Tm^{3+}$ and $Tb^{3+}$ was larger than expected for the single-Sep mutants (but not for the GB1 mutant A24Sep/K28Sep). We observed previously that the ratio between the axial tensor components of these two ions can

vary between different tags and even for the same tag at different sites of a protein (C.-T. Loh, B. Graham, E. H. Abdelkader, K. L. Tuck, G. Otting (2015) Generation of pseudocontact shifts in proteins with lanthanides using small "clickable" nitrilotriacetic acid and iminodiacetic acid tags *Chem. Eur. J.* 21, 5084-5092). These differences are not an artifact of fitting the tensors for $Tm^{3+}$ and $Tb^{3+}$ independently, as the fits yielded very similar coordinates for both metal ions. We do not understand the origin of these effects. It would help, if the effect of the ligand field could be predicted by quantum-mechanical calculations, but we were told by experts in the field that this is prohibitively difficult for lanthanide ions.

In the revised version, we propose to add the following paragraph in line 371:
"The DeltaChi tensors obtained with $Tm^{3+}$ instead of $Tb^{3+}$ ions were unexpectedly low for the single-Sep mutants, but not for the GB1 mutant A24Sep/K28Sep. We observed previously that the ratio between the DeltaChi_axial components of these two ions can vary between different tags and even for the same tag at different sites of a protein (Loh et al., 2015). These differences are not an artifact of fitting the tensors for $Tm^{3+}$ and $Tb^{3+}$ independently, as, with the exception of ubiquitin E18Sep, the fits converged to very similar metal positions (Tables 1 and 2). We do not understand the origin of different magnitudes of Chi-tensor anisotropies for $Tm^{3+}$ and $Tb^{3+}$ ions. In addition, much larger DeltaChi tensors have been reported for sterically rigid cyclen tags (Joss and Häussinger, 2019), suggesting that a rigid ligand field promotes large DeltaChi tensors."

Comment 2: The tensor for the GB1 K10D/T11Sep(Tb3+) should be reported with an axial component of -33.7 and a rhombic component of 14.7 to fulfill the axis labeling convention providing a rhombic component up to 2/3 of the axial component in absolute value. If the authors prefer to report the tensor as in Table 1, they should at least explain why. In any case, the tensor anisotropy is surprisingly large considering that the measured pcs span a range smaller than that measured for ubiquitin, and surprisingly rather rhombic. In the double phosphoserine K10Sep/T11Sep (Tb3+) mutant, the measured values of the pcs span a range which is roughly double, but the tensor is less than half with respect to that of K10D/T11Sep(Tb3+). Please, double check that no mix-up of data has occurred.

Response: Thank you for alerting us to this typo. The correct numbers for DelatChi_axial and DeltaChi_rhombic are 7.3 and 1.6, respectively.

Comment 3: Can you comment on the reason of the different sign of the tensor axial components between K10Sep/T11Sep(Tb3+) and A24Sep/K28Sep(Tb3+)? On the other hand, the sign of the axial components of Tb and Tm are usually opposite. Why are they the same in A24Sep/K28Sep.

Response: We do not understand the reason for the sign change in the tensor for Tb3+ between the K10Sep/T11Sep and A24Sep/K28Sep mutants. We double-checked and couldn't find an error. The signs were indeed wrong for the Tm3+ tensor associated with GB1 A24Sep/K28Sep(Tm3+): the correct values for the axial and rhombic components are -15.5 and -2.5, respectively.

In the revised version, we will display the isosurfaces also for $Tm^{3+}$ in Figures 2, 3 and 4 to illustrate the degree of orthogonality of the tensors between $Tm^{3+}$ and $Tb^{3+}$ (revised Figures attached).

Comment 4: Minor points: Pag.2, line 1: "As lanthanide ions display particularly large. . ." not all lanthanoids, only some of them! Pag. 2, line 2: "While paramagnetic lanthanide ions generate paramagnetic relaxation enhancements (PRE) in the protein irrespective of metal mobility" This sentence may be read that PREs do not depend on mobility, which is slightly inaccurate, because internal mobility changes the correlation time of dipole-dipole relaxation (see Fragai et al. Coord. Chem. Rev. 2013, 257, 2652 for a thorough discussion). Please, clarify this point. Caption to Fig. 3: please indicate all panel letters.

Response: In the revised version, we propose the following changes.

Page 2, line 1: "As many lanthanide ions display particularly large…"

Page 2, paragraph 2: "Paramagnetic lanthanide ions always generate paramagnetic relaxation enhancements (PRE) in the protein, which vary relatively little with minor movements of the metal ion. In contrast, PCSs can decrease dramatically if the lanthanide complex reorientates relative to the protein."

Response to the comments made by Claudio Luchinat:

We are grateful for the insightful comments and identifying errors.

Comment 1: The authors should comment that the axial anisotropies of the proposed tag attached to ubiquitin with a single phosphoserine mutation are significantly smaller than those of other previously proposed rigid tags (more than a factor 2 for Tb probes, more than a factor 5 for Tm), and should discuss the origin of this difference.

Response: The same point was picked up by Marcellus Ubbink and our response is copied here.

Indeed, the difference in DeltaChi tensors obtained with $Tm^{3+}$ and $Tb^{3+}$ was larger than expected for the single-Sep mutants (but not for the GB1 mutant A24Sep/K28Sep). We observed previously that the ratio between the axial tensor components of these two ions can vary between different tags and even for the same tag at different sites of a protein (C.-T. Loh, B. Graham, E. H. Abdelkader, K. L. Tuck, G. Otting (2015) Generation of pseudocontact shifts in proteins with lanthanides using small "clickable" nitrilotriacetic acid and iminodiacetic acid tags *Chem. Eur. J.* 21, 5084-5092). These differences are not an artifact of fitting the tensors for $Tm^{3+}$ and $Tb^{3+}$ independently, as the fits yielded very similar coordinates for both metal ions. We do not understand the origin of these effects. It would help, if the effect of the ligand field could be predicted by quantum-mechanical calculations, but we were told by experts in the field that this is prohibitively difficult for lanthanide ions.

In the revised version, we propose to add the following paragraph in line 371:
"The DeltaChi tensors obtained with $Tm^{3+}$ instead of $Tb^{3+}$ ions were unexpectedly low for the single-Sep mutants, but not for the GB1 mutant A24Sep/K28Sep. We observed previously that the ratio between the DeltaChi_axial components of these two ions can vary between different tags and even for the same tag at different sites of a protein (Loh et al., 2015). These differences are not an artifact of fitting the tensors for $Tm^{3+}$ and $Tb^{3+}$ independently, as, with the exception of ubiquitin E18Sep, the fits converged to very similar metal positions (Tables 1 and 2). We do not understand the origin of different magnitudes of Chi-tensor anisotropies for $Tm^{3+}$ and $Tb^{3+}$ ions. In addition, much larger DeltaChi tensors have been reported for sterically rigid cyclen

tags (Joss and Häussinger, 2019), suggesting that a rigid ligand field promotes large DeltaChi tensors."

Comment 2: The tensor for the GB1 K10D/T11Sep(Tb3+) should be reported with an axial component of -33.7 and a rhombic component of 14.7 to fulfill the axis labeling convention providing a rhombic component up to 2/3 of the axial component in absolute value. If the authors prefer to report the tensor as in Table 1, they should at least explain why. In any case, the tensor anisotropy is surprisingly large considering that the measured pcs span a range smaller than that measured for ubiquitin, and surprisingly rather rhombic. In the double phosphoserine K10Sep/T11Sep (Tb3+) mutant, the measured values of the pcs span a range which is roughly double, but the tensor is less than half with respect to that of K10D/T11Sep(Tb3+). Please, double check that no mix-up of data has occurred.

Response: Thank you for alerting us to this typo. The correct numbers for DelatChi_axial and DeltaChi_rhombic are 7.3 and 1.6, respectively.

Comment 3: Can you comment on the reason of the different sign of the tensor axial components between K10Sep/T11Sep(Tb3+) and A24Sep/K28Sep(Tb3+)? On the other hand, the sign of the axial components of Tb and Tm are usually opposite. Why are they the same in A24Sep/K28Sep.

Response: We do not understand the reason for the sign change in the tensor for Tb3+ between the K10Sep/T11Sep and A24Sep/K28Sep mutants. We double-checked and couldn't find an error. The signs were indeed wrong for the Tm3+ tensor associated with GB1 A24Sep/K28Sep(Tm3+): the correct values for the axial and rhombic components are -15.5 and -2.5, respectively.

In the revised version, we will display the isosurfaces also for $Tm^{3+}$ in Figures 2, 3 and 4 to illustrate the degree of orthogonality of the tensors between $Tm^{3+}$ and $Tb^{3+}$ (revised Figures attached).

Comment 4: Minor points: Pag.2, line 1: "As lanthanide ions display particularly large. . ." not all lanthanoids, only some of them! Pag. 2, line 2: "While paramagnetic lanthanide ions generate paramagnetic relaxation enhancements (PRE) in the protein irrespective of metal mobility" This sentence may be read that PREs do not depend on mobility, which is slightly inaccurate, because internal mobility changes the correlation time of dipole-dipole relaxation (see Fragai et al. Coord. Chem. Rev. 2013, 257, 2652 for a thorough discussion). Please, clarify this point. Caption to Fig. 3: please indicate all panel letters.

Response: In the revised version, we propose the following changes.

Page 2, line 1: "As many lanthanide ions display particularly large…"

[revised manuscript text omitted]

Values for the dissociation constant $K_d$ were derived from global fits to data from two and three different measurements with Tb$^{3+}$ and Tm$^{3+}$, respectively. Fits were performed either with inclusion of the binding stoichiometry $n$ as a fitting parameter or setting $n = 1$, with the result shown underneath.

| Fitted parameters | Tb$^{3+}$ | | Tm$^{3+}$ | |
|---|---|---|---|---|
| | setting $n = 1$ | fitting $n$[a] | setting $n = 1$ | fitting $n$[b] |
| ΔH (kJ mol$^{-1}$) | 15 | 23 | 20 | 12 |
| ΔS (Jmol$^{-1}$K) | 137 | 161 | 143 | 128 |
| $K_d$ (µM) | 25 | 42 | 133 | 32 |

[a] The fit yielded $n = 0.7$.
[b] The fit yielded $n = 1.4$.

[revised manuscript text omitted]

---

## Editor Decision (ED1)

**From:** Michael Sattler <michael.sattler@tum.de>
**Sent:** 06 Dec 2020 20:16
**To:** Copernicus Publications Editorial Support <editorial@copernicus.org>
**Cc:** Mark Bostock <mark.bostock@tum.de>
**Subject:** Re: mr-2020-26 (referee) - final response

…
Please find my comments enclosed, with help from Dr. Mark Bostock, a postdoc in my group.

Regards, Michael
* * *
The manuscript by Tharayil et al. demonstrates a new strategy for the use of the phosphoserine introduced via genetic code expansion using recombinant protein expression to incorporate lanthanide binding sites into proteins to enable the of paramagnetic data, i.e. pseudo-contact shifts. The authors propose that this provides an measurement efficient strategy to establish lanthanide binding sites and avoids chemical coupling with LBTs. The model systems ubiquitin and GB1 are used to assess the potential for measuring PCSs in such systems, with very low Q scores obtained, indicating excellent correlation between the experimental and back-calculated data and hence successful immobilisation of the lanthanide metal. The structural features allowing successful expression of folded protein and lanthanide binding are assessed using a range of different proteins and mutants. The authors suggest empirical guidelines for incorporation of $Ln^{3+}$ binding sites using this method, although it remains likely that there will be a considerable amount of trial and error required in this approach.

The manuscript is carefully written, the data and figures well-presented and the strengths, limitations and possible structural interpretations of the results clearly discussed. Whilst it seems likely that this technique will not be easy to implement in other systems, the manuscript provides useful information for other researchers and establishes an alternative approach for PCS determination in systems not amenable to other approaches.

**Specific comments**
− Was it always possible to achieve saturation of the various proteins with the lanthanide metals? For example, in Figure 1 (a) the complex is in slow exchange and a fraction of the peaks still appear at the (assumed) unbound position. This is also observed for other spectra in this paper e.g. Figure 3, GB1. For many applications e.g. observation of PCSs on binding partners, complete saturation with metal ions is necessary to accurately interpret the PCSs. Is this affected by the choice of metal?
− The authors say that the proteins were titrated with paramagnetic lanthanide metals. In some cases they mention that a 1:1 ratio of lanthanide:protein was used. Was this used in all cases? How did the authors avoid free Ln3+ in solution potentially creating non-specific bleaching due to the PRE component of the lanthanides ("solvent PRE"). It would be useful to comment on this.
− An impressive range of different proteins and mutants is tested in this manuscript. It would be helpful to include a supplementary table comparing all the different mutants studied in terms of number of phosphoserines, other mutations, expression level, metal binding etc.
− The rather low binding affinity for the lanthanides has potential disadvantages and would for example prohibit the use in combination with nucleic acids as free lanthanide ions will bind and potentially cleave the nucleic acid. But free lanthanides may also interfere with other regions of a given protein. The authors may want to comment and discuss this.

**Technical comments**
Line 62: "only *a* few" ("a" missing).
Line 112 (Methods): Were the lanthanide stock solutions for NMR titration also prepared in NMR buffer as for the ITC experiments?
Line 245: "The difficulties to express most of the double-phosphoserine mutants was not due to expression into insoluble inclusion bodies, as we did not find the proteins in the insoluble fraction after cell lysis." à The difficulties *in* expressing most of the double-phosphoserine mutants *were* not …. Also in line 250.

NMR spectra: The $^{15}$N axis label looks like it is divided by ppm.
Figure 2: It would be useful to provide a key to the colours on the graphs or maybe to choose different colours – blue and red could be confused with the blue and red lobes of the PCS tensors shown on the right hand side. This is true in some of the other figures too.
Figure 5: It would be helpful to mark the distances to the lanthanide metal – in particular in (a) Glu26 does not appear to be close to the metal position, whilst in (b) the proximity is evident.
Figure 6: It would be useful to mark the proposed interactions as discussed in the text.
* * *
On 03.12.2020 22:20, editorial@copernicus.org wrote:

Dear Michael Sattler,

We are pleased to inform you that the open discussion of the following MR manuscript was closed:

Title: Phosphoserine for the generation of lanthanide binding sites on proteins for paramagnetic NMR
Author(s): Sreelakshmi Mekkattu Tharayil et al.
MS No.: mr-2020-26
MS type: Research article
Special Issue: Robert Kaptein Festschrift

No more referee comments and short comments will be accepted, but the authors are encouraged to post final author comments on behalf of all co-authors (final response phase). We will inform you as soon as an author comment has been posted in the discussion forum at: https://mr.copernicus.org/preprints/mr-2020-26/#discussion

If you have prepared a review but could not meet the discussion deadline, you are kindly asked to send it directly to me at: editorial@copernicus.org

Please log in with your Copernicus Office user ID to monitor the processing of the manuscript via your MS overview at: https://editor.copernicus.org/MR/my_manuscript_overview

Thank you very much for your support and we look forward to a future cooperation. In case any questions arise, please do not hesitate to contact me.

Kind regards,

The editorial support team
Copernicus Publications
editorial@copernicus.org

---

## Author Response (AR2)

We thank the reviewer for thoughtful comments.

−    Was it always possible to achieve saturation of the various proteins with the lanthanide metals? For example, in Figure 1 (a) the complex is in slow exchange and a fraction of the peaks still appear at the (assumed) unbound position. This is also observed for other spectra in this paper e.g. Figure 3, GB1. For many applications e.g. observation of PCSs on binding partners, complete saturation with metal ions is necessary to accurately interpret the PCSs. Is this affected by the choice of metal?

Response: Indeed, hitting exact lanthanide:protein ratios in titration experiments can be quite tricky due to inaccuracies in concentration measurements (ubiquitin, for example, is devoid of a tryptophan or tyrosine residue, giving rise to low UV absorption). In addition, we experimentally observed that titration with lanthanides delivered PCSs more readily after we had treated the proteins with EDTA, but any incompletely removed traces of EDTA would compete with metal binding to the protein. We therefore adopted an operational approach to establishing the lanthanoid-protein complexes, namely by titrating until the original signal of the protein had vanished or, at least, substantially decreased (to avoid over-titration and potential binding to other sites). This is now described in lines 114-118 and 223, and discussed in a new paragraph on page 21. In addition, to avoid possible overstatements, we removed all references to exact titration ratios.
         While the paramagnetic peaks are well resolved due to PCSs, incomplete saturation with diamagnetic yttrium could indeed be problematic for signals that are significantly shifted by diamagnetic metal ions. In practice, this is not a serious problem, because PREs in the paramagnetic samples anyway prevent the observation of amide protons that are located near the metal binding site and thus most likely to shift upon titration with metals. This is now discussed on page 21. We assumed that the binding affinities of different lanthanoid ions are very similar. The close similarity in their chemical properties is well documented in the literature.

−    The authors say that the proteins were titrated with paramagnetic lanthanide metals. In some cases they mention that a 1:1 ratio of lanthanide:protein was used. Was this used in all cases? How did the authors avoid free $Ln^{3+}$ in solution potentially creating non-specific bleaching due to the PRE component of the lanthanides ("solvent PRE"). It would be useful to comment on this.

Response: We rather under-titrated the proteins with metal ions (see response to the query above). In this way we not only avoided non-specific PREs, but also the possibility of generating PCSs from alternative metal binding sites. If a significant degree of binding at other sites had occurred, it would have been manifested in a decreased quality of the fit of a single DeltaChi tensor – we detected no sign of this.

−    An impressive range of different proteins and mutants is tested in this manuscript. It would be helpful to include a supplementary table comparing all the different mutants studied in terms of number of phosphoserines, other mutations, expression level, metal binding etc.

Response: we already report all the different failed mutants in Figures S3, S6 and S8 and see no advantage in repeating this information in an additional table.

−    The rather low binding affinity for the lanthanides has potential disadvantages and would for example prohibit the use in combination with nucleic acids as free lanthanide ions will bind and potentially cleave the nucleic acid. But free lanthanides may also interfere with other regions of a given protein. The authors may want to comment and discuss this.

Response: The phosphodiester backbones of DNA and RNA are known to be sensitive to hydrolysis by free lanthanides as well as metal ions that are in complexes with remaining accessible ligand binding sites. As phosphoserine residues also change the overall charge of the protein, the lanthanide-binding strategy presented in this manuscript would quite clearly not be a good choice for studies of protein-DNA or protein-RNA complexes. We feel that discussion of these effects in the main text goes beyond the scope of the present work.

**Technical comments**
Line 62: "only *a* few" ("a" missing).

Response: fixed.

Line 112 (Methods): Were the lanthanide stock solutions for NMR titration also prepared in NMR buffer as for the ITC experiments?

Response: The stock solutions were in water (unadjusted pH = 5.7). As all protein solutions were in 20 mM HEPES buffer and the lanthanoid concentrations did not exceed 0.5 mM, any change in pH of the final solutions would have been minimal. If a change in pH had had a significant effect on the chemical shifts, this would have become apparent in worse quality factors of the DeltaChi-tensor fits.

Line 245: "The difficulties to express most of the double-phosphoserine mutants was not due to expression into insoluble inclusion bodies, as we did not find the proteins in the insoluble fraction after cell lysis." → The difficulties *in* expressing most of the double-phosphoserine mutants *were* not ….  Also in line 250.

Response: fixed.

NMR spectra: The $^{15}$N axis label looks like it is divided by ppm.

Response: that's correct. Chemical shifts are numbers with the unit ppm. Divide by the unit and the result gives the unitless numbers as they are displayed along the axes.

Figure 2: It would be useful to provide a key to the colours on the graphs or maybe to choose different colours – blue and red could be confused with the blue and red lobes of the PCS tensors shown on the right hand side. This is true in some of the other figures too.

Response: we have used this colour standard in many other articles. The figure legends state the colour code clearly.

Figure 5: It would be helpful to mark the distances to the lanthanide metal – in particular in (a) Glu26 does not appear to be close to the metal position, whilst in (b) the proximity is evident.

Response: The distance between metal ion and the nearest sidechain oxygen of Glu26 is 4.9 Å in Figure 5a and the corresponding distance to the nearest oxygen of Glu56 is 3.6 Å in Figure 5b. We now write in the main text that these glutamate side chains are potentially within reach of the metal ion (line 270). Notably, Figure 5 displays the crystal structure conformation for these glutamate residues without trying to explore alternative side chain conformations. This is now also mentioned in the figure legend. As we already stated in line 274, mutation of Glu26 to an uncharged residue abolished the observation of PCSs.

Figure 6: It would be useful to mark the proposed interactions as discussed in the text.

Response: salt bridges are rarely manifested in crystal structures by obvious proximity between the charged groups of different amino acid side chains. As part of the discussion section, the figure only means to highlight the possibility of salt bridge formation. Inspection of the full 3D structure of the proteins is required to appreciate this possibility. The figure legend gives the colour code to identify the amino acid side chains of interest.